# Formation of robust bound states of interacting microwave photons

Systems of correlated particles appear in many fields of modern science and represent some of the most intractable computational problems in nature. The computational challenge in these systems arises when interactions become comparable to other energy scales, which makes the state of each particle depend on all other particles[1]. The lack of general solutions for the three-body problem and acceptable theory for strongly correlated electrons shows that our understanding of correlated systems fades when the particle number or the interaction strength increases. One of the hallmarks of interacting systems is the formation of multiparticle bound states[2–9]. Here we develop a high-fidelity parameterizable fSim gate and implement the periodic quantum circuit of the spin-½ XXZ model in a ring of 24 superconducting qubits. We study the propagation of these excitations and observe their bound nature for up to five photons. We devise a phase-sensitive method for constructing the few-body spectrum of the bound states and extract their pseudo-charge by introducing a synthetic flux. By introducing interactions between the ring and additional qubits, we observe an unexpected resilience of the bound states to integrability breaking. This finding goes against the idea that bound states in non-integrable systems are unstable when their energies overlap with the continuum spectrum. Our work provides experimental evidence for bound states of interacting photons and discovers their stability beyond the integrability limit.

Photons that propagate in vacuum do not interact with each other; however, many technological applications and the study of fundamental physics require interacting photons. Consequently, realizing quantum platforms with strong interactions between photons constitutes a major scientific goal[10,11]. In this regard, superconducting circuits, which host excitations in the form of microwave photons, are promising candidates as they provide a configurable lattice in which a discrete number of photons can be confined to a qubit site, hop between the sites and interact with each other. The tunability of coupling elements enables photons to hop between the sites, and the non-linearity of qubits leads to interaction between the photons. The zero- and single-photon occupancies of qubits are used as the $|0\rangle$ and $|1\rangle$ states in quantum information processing. Here we also confine the dynamics to zero or single occupancy for a given qubit, the so-called hard core boson limit, and show that microwave photons can remain adjacent and form coherent bound states.

The advent of quantum processors is giving rise to a methodological shift in the studies of correlated systems[12–16]. Whereas theoretical studies of condensed matter models were focused on Hamiltonian systems for many decades, high-fidelity quantum processors commonly operate on the basis of unitary gates rather than continuous Hamiltonian dynamics. This experimental access to periodic (Floquet) unitary dynamics opens the door to a plethora of non-equilibrium phenomena[17]. Because such periodic dynamics often cannot be described in terms of a local Hamiltonian, established results are few and far between[18–20]. For instance, until recently, there was no theoretically known example of bound-state formation for interacting Floquet dynamics.

Integrable models form the cornerstone of our understanding of dynamical systems and can serve to benchmark quantum processors. A relevant example of an interacting integrable model is the one-dimensional (1D) quantum spin-½ XXZ model, which is known to support bound states[2–5,21]. Recently, the shared symmetries of the spin-½ XXZ Hamiltonian model with its Floquet counterpart led to a proof for the integrability of the XXZ Floquet quantum circuits[22–24]. Later, Aleiner obtained the full spectrum for these Floquet systems and provided analytical results for bound states[25]. The advantage of using quantum processors in studying these models becomes apparent when going beyond the integrability limit, where analytical and numerical techniques fail to scale favourably.

To define systems with bound states, consider a chain of coupled qubits and the unitary evolution $\hat{U}$ of interacting photons on this array. We divide the computational space of all bitstrings with $n_{ph}$ photons into two sets: one set $\mathcal{T}$ is composed of all bitstrings in which all photons are in adjacent sites, for example, $|00\dots011100\dots00\rangle$; the other set $\mathcal{S}$ contains all other $n_{ph}$ bitstrings, for example, $|00\dots101001\dots00\rangle$. A bound state is formed when the eigenstates of the system can be expanded as the superposition of bitstrings mainly in $\mathcal{T}$ and with smaller weight in $\mathcal{S}$. Therefore, for any initial state $|\psi_0\rangle \in \mathcal{T}$ the photons remain adjacent at all future times $|\psi\rangle = \hat{U} |\psi_0\rangle$, which implies that almost every projective measurement returns a bitstring in $\mathcal{T}$ (Fig. 1a).

The emergence of a thermodynamic phase or the formation of a bound state in Floquet dynamics seems rather implausible at first sight. In a closed Floquet system there is no notion of lowest energy, a key concept in equilibrium physics. Therefore, the energy minimization that commonly stabilizes bound states in, for example, atoms

does not hold. In the absence of interactions and in 1D, photons hop independently and the evolution can be mapped to that of free fermions. In this limit, obviously, no bound state can be formed. The key question for bound-state formation is whether the effect of kinetic energy (hopping) that moves photons away from each other could be balanced by interactions. In Fig. 1b, we provide a plausibility argument to illustrate this point. Consider two photons that are initially occupying adjacent sites, in the low kinetic energy regime in which a maximum of one hopping event occurs in the span of a few cycles. In the spirit of Feynman path formulation, the probability of a given configuration at a later time can be obtained from summing over all possible paths that lead to that configuration with proper weights. When photons are in adjacent sites, they accumulate phase due to the interaction. In the three depicted paths, the accumulated phases are different, thus leading to destructive interference. Hence, the interactions suppress the probability of unbound configurations and facilitate the formation of bound states.

The control sequence used to generate unitary evolution in our experiment consists of a periodic application of entangling gates in a 1D ring of $N_Q = 24$ qubits (Fig. 1c). Within each cycle, two-qubit fSim gates are applied between all pairs in the ring. In the two-qubit subspace, $\{|00\rangle, |01\rangle, |10\rangle, |11\rangle\}$, this gate can be written as

$$\text{fSim}(\theta, \phi, \beta) = \begin{pmatrix} 1 & 0 & 0 & 0 \\ 0 & \cos\theta & ie^{i\beta}\sin\theta & 0 \\ 0 & ie^{-i\beta}\sin\theta & \cos\theta & 0 \\ 0 & 0 & 0 & e^{i\phi} \end{pmatrix}, \quad (1)$$

where $\theta$ and $\beta$ set the amplitude and phase, respectively, of hopping between adjacent qubit lattice sites, and the conditional phase angle $\phi$ imparts a phase on the $|11\rangle$ state on interaction of two adjacent photons. In the Supplementary Information we show that we can achieve this gate with high fidelity (approximately 99%) for several angles. In the following, we will denote $\text{fSim}(\theta, \phi, \beta = 0)$ as $\text{fSim}(\theta, \phi)$. The qubit chain is periodically driven by a quantum circuit, with the cycle unitary:

$$\hat{U}_F = \prod_{\text{even bonds}} \text{fSim}(\theta, \phi, \beta) \prod_{\text{odd bonds}} \text{fSim}(\theta, \phi, \beta). \quad (2)$$

In the limit of $\beta = 0$ and $\theta, \phi \to 0$, this model becomes the Trotter–Suzuki expansion of the XXZ Hamiltonian model.

To quantify to what extent photons remain bound together, we prepare an initial state with $n_{ph}$ photons at adjacent sites and measure the photon occupancy of all sites after each cycle with approximately 5,000 repetitions. In Fig. 2a we plot the average photon occupancy $(1 - \langle \hat{Z}_j \rangle)/2$ on each site $j$ as a function of circuit depth for the fSim angles $\theta = \pi/6$ and $\phi = 2\pi/3$. Because the fSim gates are excitation number conserving, all data are postselected for the bitstrings with the proper number of excitations, which allows us to mitigate errors induced by population decay. Although $n_{ph} = 1$ is not a bound state, it provides a benchmark, where we can clearly see the quantum random walk of a single particle and its familiar interference pattern. For $n_{ph} = 2$, we observe the appearance of two wavefronts: the fastest one corresponds to unbound photons, whereas the other one corresponds to the two-photon bound state. For $n_{ph} > 2$, the concentration of the population near the centre indicates that the photons do not disperse far, but instead stay close to each other. In the Supplementary Information we also present the situation in which the initial photons are not adjacent, in which case the system tends towards a uniform distribution.

To extract the wavefront velocity, we select the measured bitstrings in which the photons remain adjacent, that is, in $\mathcal{T}$, and discard the ones in $\mathcal{S}$. In Fig. 2c, we present the spatially and temporally resolved

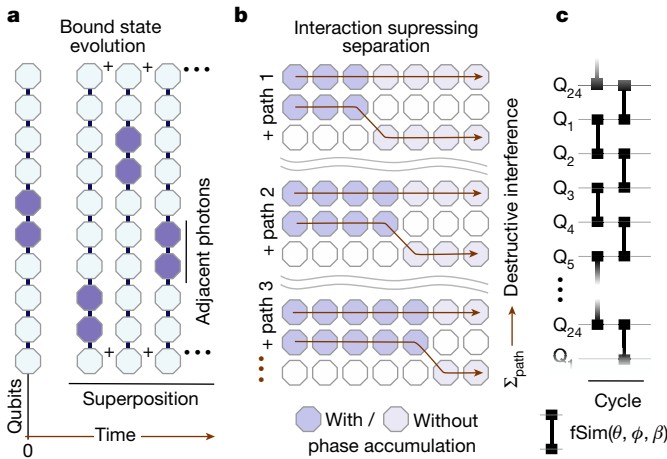

**Fig. 1 | Bound states of photons. a**, In a 1D chain of qubits hosting bound states, an initial state with adjacent photons evolves into a superposition of states in which the photons remain bound together. **b**, Interactions between photons can lead to destructive interference for paths in which photons do not stay together, thus suppressing separation. **c**, Schematic of the gate sequence used in this work. Each cycle of evolution contains two layers of fSim gates that connect the even and odd pairs, respectively. The fSim gate has three controllable parameters, which set the kinetic energy ($\theta$), the interaction strength ($\phi$) and a synthetic magnetic flux ($\beta$). The median gate infidelity, measured with cross-entropy benchmarking, is 1.1% (see Supplementary Information).

probabilities of the 'centre of photon mass' (CM, Fig. 2b) of these $\mathcal{T}$ bitstrings. With this selection, the first panel in Fig. 2c shows a very similar pattern to the single-particle propagation in Fig. 2a, highlighting the composite nature of the bound state. The propagation velocities of the bound states can now be easily seen, and, as expected, the larger bound states propagate more slowly. The wavefronts propagate with constant velocity, indicating that the bound photons move ballistically and without effects of impurity scattering. The extracted maximum group velocities of the bound states, $v_g^{max}$ (Fig. 2d), match very well with those corresponding to the analytical dispersion relations derived in ref. [25], which take the same functional form for all $n_{ph}$:

$$\cos(E(k) - \chi) = \cos^2(\alpha) - \sin^2(\alpha)\cos(k), \quad (3)$$

where $E$ is the quasi-energy, $k$ is the momentum, and $\alpha$ and $\chi$ are functions of $n_{ph}$, $\theta$ and $\phi$ (see Supplementary Information for exact forms).

To characterize the stability of the bound state, it is useful to consider the evolution of the fraction of bitstrings in which the photons remain adjacent, $n_{\mathcal{T}}/(n_{\mathcal{T}} + n_{\mathcal{S}})$ (where $n_{\mathcal{T}(\mathcal{S})}$ is the number of bitstrings in $\mathcal{T}(\mathcal{S})$), which reflects contributions from both internal unitary dynamics as well as external decoherence (Fig. 2e). In the absence of dephasing, $n_{\mathcal{T}}$ should reach a steady-state value after the observed initial drop. However, we observe a slow decay, which we attribute to the dephasing of the qubits, as the data are postselected to remove $T_1$ photon loss effects. A remarkable feature of the data is that the decay rate for various $n_{ph}$ values is the same, indicating that this decay is dominated by bond breaking at the edges of the bound state.

To show that the bound photons are quasiparticles with well-defined momentum, energy and charge, we study the spectrum of the bound states using a many-body spectroscopy technique[26]. We measure the energy of the bound states by comparing their accumulated phase over time relative to the vacuum state $|0\rangle^{\otimes N_Q}$. This is achieved by preparing $n_{ph}$ adjacent qubits in the $|+X\rangle$ state and measuring the following $n_{ph}$ body correlator that couples the bound states with the vacuum state:

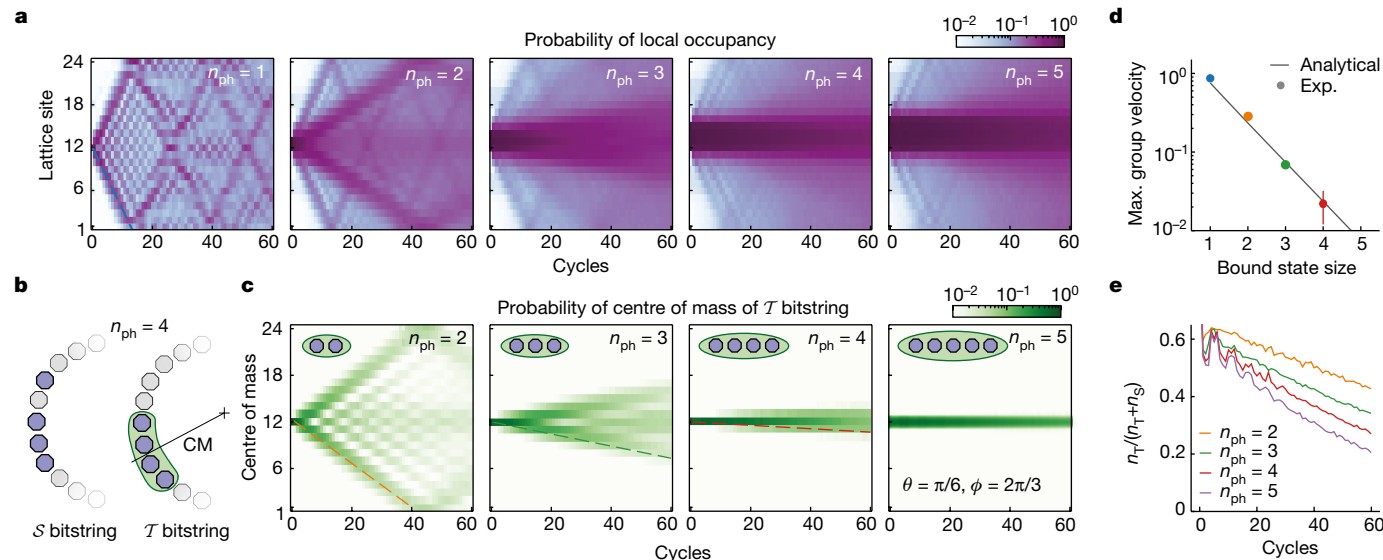

**Fig. 2 | Trajectory of bound photons. a**, Time- and site-resolved photon occupancy on a 24-qubit ring for photon numbers $n_{ph} = 1$–5. To measure a $n_{ph}$ photon bound state, $n_{ph}$ adjacent qubits are prepared in the $|1\rangle$ state. **b**, Schematic and example of bitstrings in $\mathcal{T}$ and $\mathcal{S}$. Centre of mass is defined as the centre of $n_{ph}$ adjacent occupied sites. **c**, Evolution of the centre of mass of $n_{ph}$ bound states. Each trajectory is similar to the single-photon case, highlighting the composite nature of the bound states. **d**, Extracted maximum (Max.) group velocity from the trajectory of the centre of mass. Black line, theoretical prediction. Exp., experimental. **e**, Decay of the bound state into the single excitations continuum due to dephasing. For all panels, $\theta = \pi/6$ and $\phi = 2\pi/3$, and the trajectories are averaged over all possible initial states. Data are postselected for number of excitations equal to $n_{ph}$.

$$\langle C_{j,n_{ph}}\rangle = \langle \Pi_{i=j}^{j+n_{ph}-1}\sigma_i^+\rangle = \langle \Pi_{i=j}^{j+n_{ph}-1}(X_i + iY_i)\rangle \tag{4}$$

for all sets of $n_{ph}$ adjacent qubits (Fig. 3a). This protocol is based on measuring the Green's function of the system. Whereas the correlator above is not Hermitian, it can be reconstructed by measuring its constituent terms (for example, $\langle X_j X_{j+1}\rangle - \langle Y_j Y_{j+1}\rangle + i\langle X_j Y_{j+1}\rangle + i\langle Y_j X_{j+1}\rangle$ for $n_{ph} = 2$) and summing these with the proper complex prefactors. We note that as $C_{j,n_{ph}}$ only couples the $n_{ph}$ photon terms to the vacuum, the initial product state used here serves the same purpose as an entangled superposition state $|000..00\rangle + |00..0110..00\rangle$. By expanding these states in the momentum basis ($k$-space), it becomes evident that $\langle C_{j,n_{ph}}\rangle$ contains the phase information needed to evaluate the dispersion relation of the $n_{ph}$ bound states:

$$\begin{aligned}|\psi(t)\rangle &= \frac{1}{\sqrt{2}}\left(|0\rangle^{\otimes N_Q} + \sum_k \alpha_k e^{-i\omega(k)t}|k\rangle\right) \\ &\rightarrow \langle C_{j,n_{ph}}\rangle = 1/(2\sqrt{N_Q})\sum_k \alpha_k^* e^{i(\omega(k)t - kj)},\end{aligned} \tag{5}$$

where $|k\rangle$ and $\alpha_k$ are bound $n_{ph}$ photon momentum states and their coefficients, respectively.

Figure 3b shows the real and imaginary parts of the correlator for the case of two photons. Whereas the real-space data display a rather intricate pattern (Fig. 3b), conversion to the energy and momentum domain through a two-dimensional (2D) Fourier transform reveals a clear band structure for both the single-particle and the many-body states (Fig. 3c). The observed bands, which are defined modulo $2\pi$ per cycle due to the discrete time translation symmetry of the Floquet circuit, are in agreement with the predictions of equation (3), as illustrated in coloured dashed curves. The bands shift when the photon number increases, as expected from the higher total interaction energy. Moreover, they become flatter, a characteristic feature of increased interaction effects.

For a bound state to form, the interaction energy must be sufficiently high compared to the kinetic energy of the particles. In particular, bound states are only expected to exist for all momenta when $\phi > 2\theta$ (ref. [25]).

To explore this dependence on $\phi/\theta$, we also measure the band structure for $n_{ph} = 2$ in the weakly interacting regime ($\theta = \pi/3$, $\phi = \pi/6$; Fig. 3d), which exhibits very different behaviour from the more strongly interacting case studied in Fig. 3c: although no band is observed for most momenta, a clear state emerges near $k = \pm\pi$ per site. Interestingly, this observation of a bound state in the weakly interacting regime can be attributed to destructive interference of the decay products of the bound state: a two-photon bound state $|..0110..\rangle$ can separate into two possible states, $|..1010..\rangle$ and $|..0101..\rangle$, which are shifted relative to each other by one lattice site. Hence, they destructively interfere when the momentum is near $k = \pm\pi$ per site, which prevents separation. (See the Supplementary Information for band structures of additional fSim angles.)

External magnetic fields can shift the energy bands and reveal the electric pseudo-charge of the quasiparticles constituting the band. We produce a synthetic magnetic flux $\mathbf{\Phi}$ that threads the ring of qubits by performing $Z$ rotations with angles $\pm\mathbf{\Phi}/N_Q$ on the qubits before and after the two-qubit fSim gates, resulting in a complex hopping phase $\beta = \mathbf{\Phi}/N_Q$ when a photon moves from site $j$ to $j+1$ (ref. [27]). As a consequence, the eigenstates are expected to attain a phase $j(n_{ph}\beta)$, effectively shifting their quasi-momentum by $n_{ph}\beta$. Figure 3e displays the flux dependence of the two-photon band structure, exhibiting a clear shift in momentum as $\mathbf{\Phi}$ increases. In Fig. 3f, we extract the shift for $n_{ph} = 1$–5 and observe excellent agreement with the theoretical predictions[25]. Crucially, the momentum shift is found to scale linearly with $n_{ph}$, indicating that the observed states have the correct pseudo-charge.

Generally, bound states in the continuum are rare and very fragile, and their stability relies on integrability or symmetries[28,29]. Familiar stable dimers, such as excitons in semiconductors, have energy resonances in the spectral gap. In the system considered here, the bound states are predicted to almost always be inside the continuum due to the periodicity of the quasi-energy. Our results shown in Fig. 3 demonstrate an experimental verification of this remarkable theoretical prediction in the integrable limit and constitute our first major result.

Next we probe the stability of the bound states against integrability breaking. Fermi's golden rule suggests that any weak perturbation that breaks the underlying symmetry will lead to an instability and a rapid

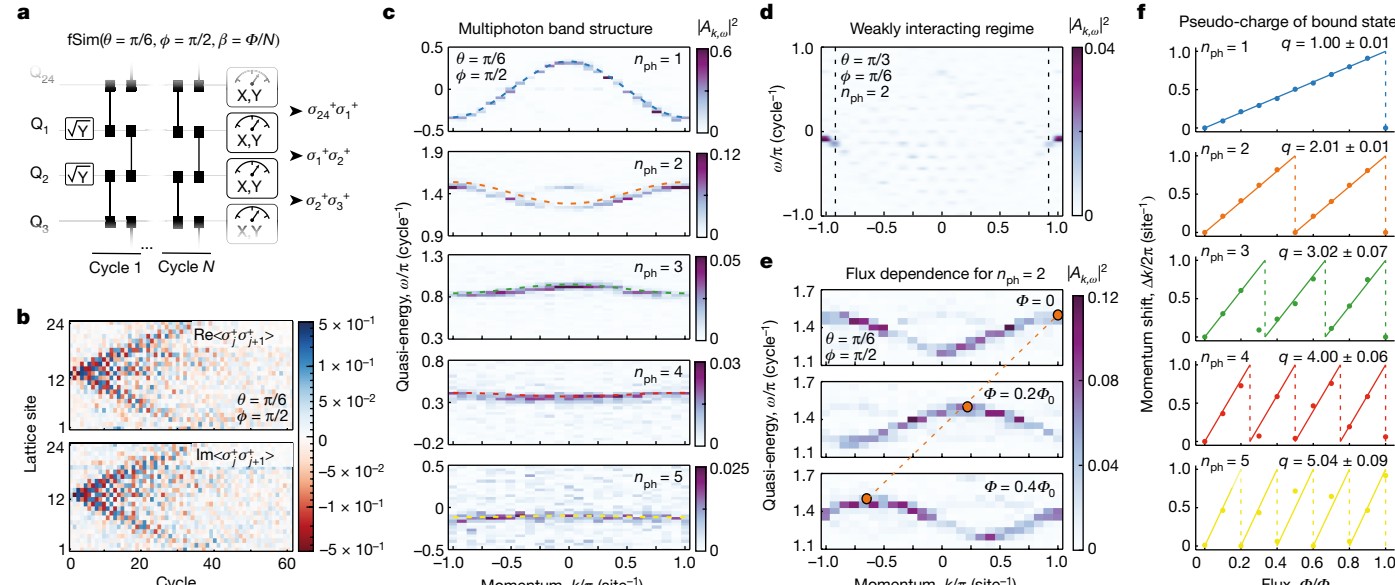

**Fig. 3 | Band structure of multiphoton bound states. a**, Schematic of circuit used for many-body spectroscopy. $n_{ph}$ adjacent qubits are prepared in the $|+\rangle$ state, before evolving the state with a variable number of fSim gates. The phase of the bound state is probed by measuring the correlator $\langle \sigma_i^+ .. \sigma_{i+n_{ph}-1}^+ \rangle$ for all sets of $n_{ph}$ adjacent qubits. **b**, Real (top, Re) and imaginary (bottom, Im) parts of the $n_{ph} = 2$ correlator. **c**, Band structure for $n_{ph} = 1-5$ (top to bottom), obtained via a 2D Fourier transform in space and time of the $n_{ph}$ correlators. Colour scale: absolute square of the Fourier transform, $|A_{k,\omega}|^2$. Dashed curves: theoretical prediction in equation (3). **d**, Band structure for $n_{ph} = 2$ in the weakly interacting ($\phi < 2\theta$) regime, displaying the emergence of a bound state only at momenta

near $k = \pm\pi$. Dashed black lines: theoretically predicted momentum threshold for the existence of the bound state (see Supplementary Information). **e**, Flux dependence of the $n_{ph} = 2$ band structure, displaying a gradual momentum shift as the flux increases ($\Phi_0 = 2\pi N_Q$). Orange circles and dashed line indicate the peak position of the band. **f**, Extracted momentum shifts as a function of flux for $n_{ph} = 1-5$ (top to bottom), indicating that the rate of shifting scales linearly with the photon number of the bound states, that is, the pseudo-charge $q$ of each bound state is proportional to its number of photons. Coloured lines: theoretical prediction.

decay of the bound states into the continuum. We examine the robustness of the $n_{ph} = 3$ bound state by constructing a quasi-1D lattice in which every other site of the 14-qubit ring is coupled to an extra qubit site (Fig. 4a). The extra sites increase the Hilbert space dimension and ensure that the system is not integrable. We implement the circuit

depicted in Fig. 4b with three layers of fSim gates in each cycle. The first two layers are the XXZ ring dynamics with the same parameters as used in Fig. 2: $\theta = \pi/6$ and $\phi = 2\pi/3$. In the third layer we also use $\phi' = 2\pi/3$ but vary the swap angle $\theta'$ to tune the strength of the integrability breaking perturbation.

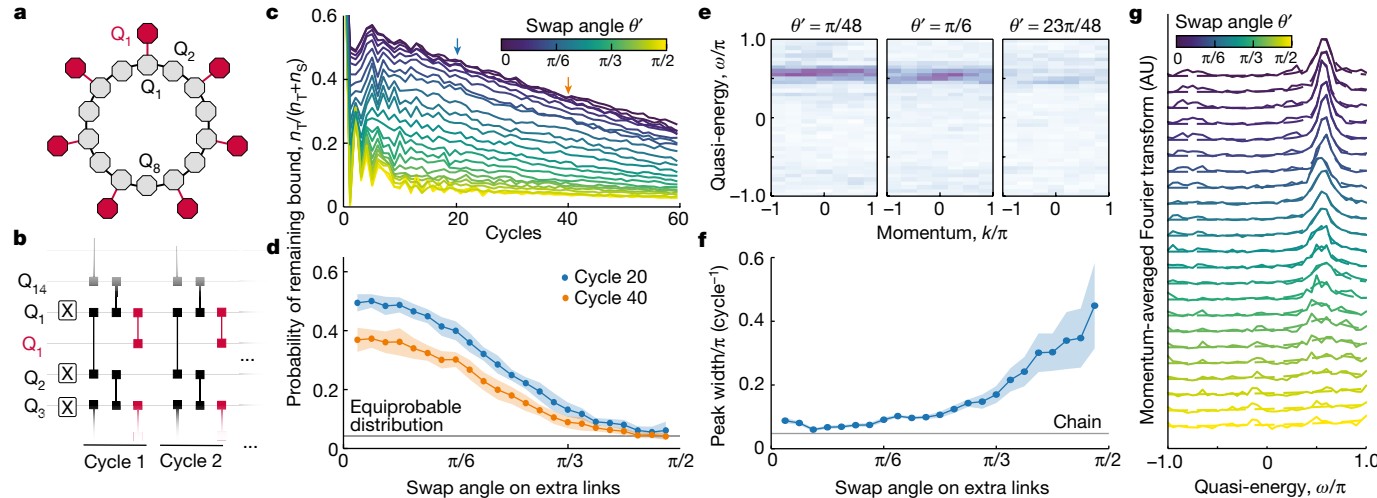

**Fig. 4 | Resilience to integrability breaking. a**, Schematic of the 14-qubit chain with seven extra sites in red to break the integrability. **b**, Integrability is broken via an extra layer of fSim gates (red) between the chain and the extra qubits, with $\phi' = \phi$ and a gradually varied $\theta'$. **c**, Decaying probability of remaining bound for different swap angles $\theta'$. Similar to Fig. 2e, the bound state decays into the continuum due to the dephasing. **d**, Probability of remaining bound after 20 and 40 cycles as $\theta'$ is swept. **e**, Spectroscopy of the $n_{ph} = 3$ bound

state for different $\theta'$. Note that the bound state survives even for $\theta' = \theta$. **f**, Half-width of the momentum-averaged spectra (from **g**) as a function of $\theta'$. The grey line indicates the result for the chain without the extra qubits. **g**, Momentum-averaged quasi-energy spectra for varying $\theta'$ fitted with Lorentzian. The bound state peak slowly disappears with the increase of $\theta'$. AU, arbitrary units.

# Article

Figure 4c shows the probability of measuring three-photon $\mathcal{T}$-bitstrings as a function of time for various $\theta'$ angles. In the limit of small $|\theta'|$, for which the integrability breaking is weak, the system shows a slowly decaying probability, similar to the unperturbed (integrable, $\theta' = 0$) results presented in Fig. 2. In Fig. 4d, we show the dependence of this probability on perturbation strength after two fixed circuit depths. For strong perturbations, the integrability breaking washes out the bound state and the probability rapidly decays to the equiprobable distribution in the full Hilbert space of three photons ($\mathcal{T}+\mathcal{S}$). However, the surprising finding is that even up to $\theta' = \pi/6$, which corresponds to perturbation gates identical to the gates on the main ring, that is, a strong perturbation, there is very little decay in $n_{\mathcal{T}}$. This observation demonstrates the resilience of the bound state to perturbations far beyond weak integrability breaking for $n_{\mathrm{ph}} = 3$. We further confirm this finding by performing spectroscopy of these states, which shows the presence of the $n_{\mathrm{ph}} = 3$ bound states up to large perturbations (Fig. 4e). By fitting the momentum-averaged spectra (Fig. 4g), we extract the $\theta'$ dependence of the half-width of the band (Fig. 4f). Indeed, we find that the bandwidth is insensitive to $\theta'$ up to very large perturbation.

These observations are at odds with the expectation that nonintegrable perturbation leads to the fast decay of bound states into the continuum. One known exception is many-body scars, in which certain initial states exhibit periodic revivals and do not thermalize[30,31]. Moreover, in the case of weak integrability breaking, robustness to perturbations can result from quasi-conserved or hidden conserved quantities[32,33]. However, the resilience observed here extends well beyond the weak integrability breaking regime typically considered in such scenarios[34]. Alternatively, the presence of highly incommensurate energy scales in the problem can lead to a very slow decay in a chaotic system due to parametrically small transition matrix elements, a phenomenon called prethermalization[35,36]. Our experiment finds the survival of an integrable system's feature—bound states—for large perturbation and in the absence of obvious scale separation, which may point to a new regime arising due to interplay of integrability and prethermalization.

The key enabler of our experiment is the capability of tuning high-fidelity fSim gates to change the ratio of kinetic to interaction energy, as well as directly measuring multibody correlators $\langle C_{j,n_{\mathrm{ph}}}\rangle$, both of which are hard to access in conventional solid-state and atomic physics experiments. Aided by these capabilities, we observed the formation of multiphoton bound states and discovered a striking resilience to non-integrable perturbations. This experimental finding, although still observed for computationally tractable scales, in the absence of any theoretical prediction, constitutes our second major result (Fig. 4). A proper understanding of this unexpected discovery is currently lacking.

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

A. Morvan[1,8], T. I. Andersen[1,8], X. Mi[1,8], C. Neill[1,8], A. Petukhov[1], K. Kechedzhi[1], D. A. Abanin[1,2], A. Michailidis[2], R. Acharya[1], F. Arute[1], K. Arya[1], A. Asfaw[1], J. Atalaya[1], J. C. Bardin[1,3], J. Basso[1], A. Bengtsson[1], G. Bortoli[1], A. Bourassa[1], J. Bovaird[1], L. Brill[1], M. Broughton[1], B. B. Buckley[1], D. A. Buell[1], T. Burger[1], B. Burkett[1], N. Bushnell[1], Z. Chen[1], B. Chiaro[1], R. Collins[1], P. Conner[1], W. Courtney[1], A. L. Crook[1], B. Curtin[1], D. M. Debroy[1], A. Del Toro Barba[1], S. Demura[1],

A. Dunsworth[1], D. Eppens[1], C. Erickson[1], L. Faoro[1], E. Farhi[1], R. Fatemi[1], L. Flores Burgos[1], E. Forati[1], A. G. Fowler[1], B. Foxen[1], W. Giang[1], C. Gidney[1], D. Gilboa[1], M. Giustina[1], A. Grajales Dau[1], J. A. Gross[1], S. Habegger[1], M. C. Hamilton[1], M. P. Harrigan[1], S. D. Harrington[1], M. Hoffmann[1], S. Hong[1], T. Huang[1], A. Huff[1], W. J. Huggins[1], S. V. Isakov[1], J. Iveland[1], E. Jeffrey[1], Z. Jiang[1], C. Jones[1], P. Juhas[1], D. Kafri[1], T. Khattar[1], M. Khezri[1], M. Kieferová[1,4], S. Kim[1], A. Y. Kitaev[1,5], P. V. Klimov[1], A. R. Klots[1], A. N. Korotkov[1,6], F. Kostritsa[1], J. M. Kreikebaum[1], D. Landhuis[1], P. Laptev[1], K.-M. Lau[1], L. Laws[1], J. Lee[1], K. W. Lee[1], B. J. Lester[1], A. T. Lill[1], W. Liu[1], A. Locharla[1], F. Malone[1], O. Martin[1], J. R. McClean[1], M. McEwen[1,7], B. Meurer Costa[1], K. C. Miao[1], M. Mohseni[1], S. Montazeri[1], E. Mount[1], W. Mruczkiewicz[1], O. Naaman[1], M. Neeley[1], A. Nersisyan[1], M. Newman[1], A. Nguyen[1], M. Nguyen[1], M. Y. Niu[1], T. E. O'Brien[1], R. Olenewa[1], A. Opremcak[1], R. Potter[1], C. Quintana[1], N. C. Rubin[1], N. Saei[1], D. Sank[1], K. Sankaragomathi[1], K. J. Satzinger[1], H. F. Schurkus[1], C. Schuster[1], M. J. Shearn[1], A. Shorter[1], V. Shvarts[1], J. Skruzny[1], W. C. Smith[1], D. Strain[1], G. Sterling[1], Y. Su[1], M. Szalay[1], A. Torres[1], G. Vidal[1], B. Villalonga[1], C. Vollgraff-Heidweiller[1], T. White[1], C. Xing[1], Z. Yao[1], P. Yeh[1], J. Yoo[1], A. Zalcman[1], Y. Zhang[1], N. Zhu[1], H. Neven[1], D. Bacon[1], J. Hilton[1], E. Lucero[1], R. Babbush[1], S. Boixo[1], A. Megrant[1], J. Kelly[1], Y. Chen[1], V. Smelyanskiy[1], I. Aleiner[1✉], L. B. Ioffe[1✉] & P. Roushan[1✉]

[1]Google Research, Mountain View, CA, USA. [2]Department of Theoretical Physics, University of Geneva, Geneva, Switzerland. [3]Department of Electrical and Computer Engineering, University of Massachusetts, Amherst, MA, USA. [4]Centre for Quantum Computation and Communication Technology, Centre for Quantum Software and Information, Faculty of Engineering and Information Technology, University of Technology Sydney, Sydney, New South Wales, Australia. [5]Institute for Quantum Information and Matter, California Institute of Technology, Pasadena, CA, USA. [6]Department of Electrical and Computer Engineering, University of California, Riverside, CA, USA. [7]Department of Physics, University of California, Santa Barbara, CA, USA. [8]These authors contributed equally: A. Morvan, T. I. Andersen, X. Mi, C. Neill. ✉e-mail: igoraleiner@google.com; ioffel@google.com; pedramr@google.com

## Data availability

The datasets generated and analysed for this study are available at https://doi.org/10.5281/zenodo.6981407.

**Acknowledgements** A. Morvan and P.R. acknowledge fruitful discussion with M. Devoret.

**Author contributions** A. Morvan, C.N., I.A., L.B.I. and P.R. designed the experiment. A. Morvan and T.I.A. performed the experiment and analysed the data and wrote the supplement. K.K., D.A.A., I.A. and L.B.I. provided theoretical support and analysis. A. Morvan, T.A., X.M., C.N. and A.P. developed the calibration of the fSim gate. D.A.A. and A. Michailidis performed numerical simulation in the supplements. A. Morvan, T.A., I.A., L.B. and P.R. wrote the manuscript. All authors contributed to revising the manuscript and the Supplementary Information. All authors contributed to the experimental and theoretical infrastructure to enable the experiment.

**Competing interests** The authors declare no competing interests.

## Additional information

**Correspondence and requests for materials** should be addressed to I. Aleiner, L. B. Ioffe or P. Roushan.

