## [Peer Review File · Nature]

Manuscript Title: Formation of robust bound states of interacting microwave photons

Reviewer Comments & Author Rebuttals

Reviewer Reports on the Initial Version:

Referees' comments:

Referee #1 (Remarks to the Author):

I was really excited to read this paper! It is like living in the theoretician's dream seeing a quantum computer simulation of simple toy modes with fragile but beautiful mathematical features, such as integrability. The paper reports on the simulation of XXZ Floquet circuit and studies robustness of bound states of magnons, theoretically predicted from integrability techniques, against both dephasing noise and integrability breaking. It shows that in both cases, bound states are rather robust.

I would be happy to recommend the paper for publication in Nature.

I have, however, a couple of remarks on the manuscript that I would like the authors to consider.

- the phrase "classical counterpart of the circuit shows chaos" is confusing. What is really meant there? Should one think of a classical limit of large spin quantum number or something else? If so, why this classical chaos should be relevant for $s=1/2$ (qubit) regime? perhaps it is enough to refer to "quantum chaos" as characterised e.g. by level statistics (SFF etc).

- I am really not very comfortable with the phrase "bound state of photons". Why don't author use a more "neutral" description such as a "bound state of qubits"?

- it is not clear to me why the authors have to change topology of the chain in order to break integrability?

They could also take a non-integrable (non 6-vertex) variant of fSIM gate and make the chain homogeneous (like the integrable XXZ case). Would such integrability breaking be less robust as the one discussed in the paper?

- The authors study only states with rather small number of excitation (n_{ph}). One could say they focus on "zero-temperature" physics, studying fundamental quasi-particle excitations in the model. What happens if one would try to approach the "half-filled", or at least multiple quasi-particle sectors? Could quantum simulations still be made in a controllable way? Could one measure spin-spin correlation functions in states with large n_{ph} ?

- As for the references I have two remarks:

a) The first paper which proposed the underlying mechanism for the integrability of such unitary

Floquet quantum circuits, based on the unitary representation of the R-matrix solution of the Yang-Baxter equation, was Phys. Rev. Lett. 121, 030606 (2018). I suggest the authors add this reference, together with 21,22.

b) The authors are excited about robustness of Floquet quantum dynamics to integrability breaking. This was exactly the excitement motivating numerical simulations of one of the first many-body quantum Floquet studies quarter of century ago: Phys. Rev. Lett. 80, 1808 (1998). Perhaps this work could be mentioned.

Referee #2 (Remarks to the Author):

The authors defined a 24 qubit simulator to investigate formation of real-space bound states of excitations. I think this is a very original research which should be published before the authors would answer the questions outlined below. I have no doubts about experimental side of this work, however theoretical interpretation is not so obvious to me.

One of the main statements, related to the theory side in the paper, is that the integrable Floquet protocol from the paper by Aleiner, <https://arxiv.org/pdf/2107.05715.pdf> was implemented here. I have serious suspicions that it is not actually integrable in the traditional sense. First of all, according to several existing definitions, see e.g. <https://scipost.org/10.21468/SciPostPhys.2.3.021>, <https://journals.aps.org/prl/abstract/10.1103/PhysRevLett.121.030606> or a recent review in <https://arxiv.org/abs/2206.15142>

the local unitary two-qubit gates should satisfy a Yang-Baxter equation. This is not the case in the present paper, even for $\beta=0$, and also in the paper from Aleiner. It is not the R matrix of a 6-vertex model (which is equivalent to the XXZ chain) as claimed in the manuscript. In the latter case the 11 element of the 4×4 R-matrix (which is U in the present case) must be equal to the 44 element. The 44 element should not be given by a phase factor $\exp(i\phi)$. I can confirm that with a simple check with e.g. Mathematica - this matrix is not a solution of the Yang-Baxter equation. I found also that eq. (21) in the paper by Aleiner is an exponential of a XXZ Hamiltonian with a MAGNETIC FIELD, which does not make it integrable automatically, explicitly:

$$\text{Eq (21)} = \exp \left(i \alpha (\sigma^+_{1} \sigma^-_{2} + \sigma^-_{1} \sigma^+_{2}) + i \phi / 2 (\sigma^z_{1} \sigma^z_{2} + 1) - i \phi / 2 (\sigma^z_{1} + \sigma^z_{2}) \right),$$

however I do not exclude solvability (but not in the the Yang-Baxter sense) - we know that the XXZ chain + magnetic field in Z-direction is still integrable. However, to be interpreted in this way the Trotter step should be made very short to insure integrability. I would suggest an addition to the supplement which demonstrates this.

So, I would like to see clear up the clarification of the relevance of the protocol to the XXZ model

(+the field possibly) and its integrability. The XXZ model has a SINGLE parameter, namely the anisotropy. It is not clear how this one is related to the TWO parameters of the U matrix (θ and ϕ) in the paper. I therefore suspect this story is about a XXZ chain+field.

My feeling is that the theoretical interpretation of the experimental results should be reconsidered a bit. I do think they are related to integrability but should be re-interpreted creatively.

I will be happy to re-review this paper again once the suggestions above will be addressed properly.

Referee #3 (Remarks to the Author):

The authors present a series of results from two devices with superconducting qubits in a ring geometry. They use stroboscopic application of gates to engineer an effective spin-1/2 XXZ model with tunable parameters controlled by the phase angles of the gates.

They make use of site-resolved parallel readout and tunability of the parameters of the applied gates to characterize the response of the XXZ model and examine multiparticle bound states of the model. They examine the time dynamics of initial states in which sets of neighboring qubits are excited and show binding of these groups of excitations. They also implement measurement of a multiparticle correlator which they use to extract the eigenenergy spectrum of the model for different parameters.

They demonstrate the existence of bound states in the continuum and their robustness to integrability breaking perturbations using a second device with a modified quasi-1d ring geometry.

The manuscript is clearly written with detailed and informative figures, and the result presented explores a new regime of precision simulation of few-body physics, and makes novel use of a measurement of a many-body correlator to extract spectral information. I recommend publication.

Minor typos:

Fig1 Caption: "The median gate fidelity... is 1.1%". Presumably this is 1% gate infidelity.

Page 2, second column, top paragraph: "we can achieve this gate with high fidelity (~1%). Presumably this is 99% fidelity.

Fig 4 c,d: Y axis label is. "Probability of remaining bounded". I think that this should probably be probability of remaining bound.

Author Rebuttals to Initial Comments:

Response to referees

Referee #1 :

I was really excited to read this paper! It is like living in the theoretician's dream seeing a quantum computer simulation of simple toy modes with fragile but beautiful mathematical features, such as integrability. The paper reports on the simulation of XXZ Floquet circuit and studies robustness of bound states of magnons, theoretically predicted from integrability techniques, against both dephasing noise and integrability breaking. It shows that in both cases, bound states are rather robust.

I would be happy to recommend the paper for publication in Nature.

We are really happy to see the reviewer's enthusiasm about our manuscript and greatly appreciate his/her valuable comments. We thank the reviewer for taking the time to review our manuscript and address the comments below.

I have, however, a couple of remarks on the manuscript that I would like the authors to consider.

- the phrase "classical counterpart of the circuit shows chaos" is confusing. What is really meant there? Should one think of a classical limit of large spin quantum number or something else? If so, why this classical chaos should be relevant for $s=1/2$ (qubit) regime? perhaps it is enough to refer to "quantum chaos" as characterized e.g. by level statistics (SFF etc).

We agree with the reviewer that the phrase was confusing. We have simplified the sentence to avoid the confusion.

- I am really not very comfortable with the phrase "bound state of photons". Why don't author use a more "neutral" description such as a "bound state of qubits"?

We understand the issue raised by the reviewer. However, we think that "photon" is a correct description of the physics taking place in our hardware: the excitations in the transmon qubits are indeed microwave photons, and we note that this applies even when a nonlinearity is introduced to restrict the dynamics to the first excited state. Nevertheless, we fully agree with the reviewer that it is valuable to clarify this to the reader, and have therefore addressed his/her concern with the following modification in the first paragraph: "In this regard, superconducting circuits, which host excitations in the form of localized microwave photons, are promising candidates since they provide a configurable lattice where a discrete number of photons can be confined to a qubit site, hop between the sites, and interact with each other."

- it is not clear to me why the authors have to change topology of the chain in order to break integrability?

They could also take a non-integrable (non 6-vertex) variant of fSIM gate and make the chain homogeneous (like the integrable XXZ case). Would such integrability breaking be less robust as the one discussed in the paper?

The reviewer here raises a very interesting point. Indeed, during the preliminary phase of the work, we tried several different integrability breaking schemes to see the decay of the bound state into the continuum. Importantly we observed very strong robustness in all the studied cases. The change of the topology was for us an explicit technical challenge as it required more cycles (3 cycles of fSim gates instead of 2 for the chain) and pushed the limit of our hardware further. Moreover, the extra site increases the Hilbert size much faster than the number of integrals of motion, thus making it a very strong way to break integrability.

We have added a discussion of this integrability breaking scheme in the supplementary, including explicit evidence that integrability is broken with the change of the topology.

- The authors study only states with rather small number of excitation (n_{ph}). One could say they focus on “zero-temperature” physics, studying fundamental quasi-particle excitations in the model. What happens if one would try to approach the “half-filled”, or at least multiple quasi-particle sectors? Could quantum simulations still be made in a controllable way? Could one measure spin-spin correlation functions in states with large n_{ph} ?

We completely agree with the reviewer that it could be very interesting to study states with a higher number of photons (relative to the chain size), and this will very likely be the subject of future studies. One of the main technical challenges with increasing the photon number is the post-selection needed. As the number of photons increases, the postselection becomes more strict and the signal-to-noise ratio decreases exponentially.

Also, since large bound states have very small group velocity, a proper study would require much greater circuit depths than we can currently achieve. We are actively working on these areas since we believe this will be a very important step for this type of simulation.

- As for the references I have two remarks:

a) The first paper which proposed the underlying mechanism for the integrability of such unitary Floquet quantum circuits, based on the unitary representation of the R-matrix solution of the Yang-Baxter equation, was Phys. Rev. Lett. 121, 030606 (2018). I suggest the authors add this reference, together with 21,22.

b) The authors are excited about robustness of Floquet quantum dynamics to integrability breaking. This was exactly the excitement motivating numerical simulations of one of the first many-body quantum Floquet studies quarter of century ago: Phys. Rev. Lett. 80, 1808 (1998). Perhaps this work could be mentioned.

We have added the references suggested by the referee, and thank them for this great suggestion.

Referee #2 :

The authors defined a 24 qubit simulator to investigate formation of real-space bound states of excitations. I think this is a very original research which should be published before the authors would answer the questions outlined below. I have no doubts about experimental side of this work, however theoretical interpretation is not so obvious to me.

We thank the reviewer for reviewing our work and address their comments about the theoretical interpretation below.

One of the main statements, related to the theory side in the paper, is that the integrable Floquet protocol from the paper by Aleiner, <https://arxiv.org/pdf/2107.05715.pdf> was implemented here. I have serious suspicions that it is not actually integrable in the traditional sense. First of all, according to several existing definitions, see e.g. <https://scipost.org/10.21468/SciPostPhys.2.3.021>, <https://journals.aps.org/prl/abstract/10.1103/PhysRevLett.121.030606> or a recent review in <https://arxiv.org/abs/2206.15142>

the local unitary two-qubit gates should satisfy a Yang-Baxter equation. This is not the case in the present paper, even for $\beta=0$, and also in the paper from Aleiner. It is not the R matrix of a 6-vertex model (which is equivalent to the XXZ chain) as claimed in the manuscript. In the latter case the 11 element of the 4×4 R-matrix (which is U in the present case) must be equal to the 44 element. The 44 element should not be given by a phase factor $\exp(i\phi)$. I can confirm that with a simple check with e.g. Mathematica - this matrix is not a solution of the Yang-Baxter equation. I found also that eq. (21) in the paper by Aleiner is an exponential of a XXZ Hamiltonian with a MAGNETIC FIELD, which does not make it integrable automatically, explicitly:

$$\text{Eq (21)} = \exp \left(i \alpha (\sigma^+_{-1} \sigma^-_{-2} + \sigma^-_{-1} \sigma^+_{-2}) + i \phi / 2 (\sigma^z_{-1} \sigma^z_{-2} + 1) - i \phi / 2 (\sigma^z_{-1} + \sigma^z_{-2}) \right),$$

however I do not exclude solvability (but not in the Yang-Baxter sense) - we know that the XXZ chain + magnetic field in Z-direction is still integrable. However, to be interpreted in this way the Trotter step should be made very short to insure integrability. I would suggest an addition to the supplement which demonstrates this.

We understand that the way this was presented could cause confusion, and thank the referee for bringing this to our attention. Importantly, we would like to point out that the Floquet XXZ model was shown to be integrable by Marko Ljubotina, Lenart Zadnik and Tomaz Prosen, [Phys. Rev. Lett. 122, 150605 \(2019\)](https://arxiv.org/abs/1905.0605): see appendix B and C where the authors clearly state that the R-matrix satisfies the Yang-Baxter equation. To eliminate any further confusion, we have added a supplement section (see Supplements Section VI) where we rederive the Yang-Baxter equation and provide some insight into its derivation.

So, I would like to see clear up the clarification of the relevance of the protocol to the XXZ model (+the field possibly) and its integrability. The XXZ model has a SINGLE parameter, namely the

anisotropy. It is not clear how this one is related to the TWO parameters of the U matrix (θ and ϕ) in the paper. I therefore suspect this story is about a XXZ chain+field.

My feeling is that the theoretical interpretation of the experimental results should be reconsidered a bit. I do think they are related to integrability but should be re-interpreted creatively.

I will be happy to re-review this paper again once the suggestions above will be addressed properly.

We hope that we have clarified the theoretical point raised by the referee, and would like to thank them for motivating the addition of the new theory section in the supplementary materials.

Referee #3:

The authors present a series of results from two devices with superconducting qubits in a ring geometry. They use stroboscopic application of gates to engineer an effective spin-1/2 XXZ model with tunable parameters controlled by the phase angles of the gates.

They make use of site-resolved parallel readout and tunability of the parameters of the applied gates to characterize the response of the XXZ model and examine multiparticle bound states of the model. They examine the time dynamics of initial states in which sets of neighboring qubits are excited and show binding of these groups of excitations. They also implement measurement of a multiparticle correlator which they use to extract the eigenenergy spectrum of the model for different parameters.

They demonstrate the existence of bound states in the continuum and their robustness to integrability breaking perturbations using a second device with a modified quasi-1d ring geometry.

The manuscript is clearly written with detailed and informative figures, and the result presented explores a new regime of precision simulation of few-body physics, and makes novel use of a measurement of a many-body correlator to extract spectral information. I recommend publication.

We thank the reviewer for the review and the very positive comments about our manuscript. We would like to point out that all the experiments in the manuscript were performed on the same device, leveraging the square lattice connectivity of our quantum hardware.

Minor typos:

Fig1 Caption: "The median gate fidelity... is 1.1%". Presumably this is 1% gate infidelity.

Page 2, second column, top paragraph: "we can achieve this gate with high fidelity (~1%). Presumably this is 99% fidelity.

Fig 4 c,d: Y axis label is. "Probability of remaining bounded". I think that this should probably be probability of remaining bound.

We have corrected the typos pointed out by the referee and thank them for bringing these to our attention.

Reviewer Reports on the First Revision:

Referees' comments:

Referee #1 (Remarks to the Author):

I am happy with the revised version of the manuscript, the authors have implemented or adequately answered all the points I have raised. Therefore, I recommend publication of the manuscript in the current form.

Referee #2 (Remarks to the Author):

I agree with the revised version of the main text and with improvements and additions made in the Supplement. I am happy to recommend this paper for publication without further delay.